# The Quality of Life of Children with Myelomeningocele: A Cross-Sectional Preliminary Study

**DOI:** 10.3390/ijerph182010756

**Published:** 2021-10-13

**Authors:** Anna Rozensztrauch, Magdalena Iwańska, Maciej Bagłaj

**Affiliations:** 1Department of Pediatrics, Division of Neonatology, Wroclaw Medical University, 50-367 Wrocław, Poland; magdalena2.iwanska@gmail.com; 2Department of Pediatrics, Division of Propaedeutic of Pediatrics and Rare Disorders, Wroclaw Medical University, 50-367 Wrocław, Poland; maciej.baglaj@umed.wroc.pl

**Keywords:** quality of life, myelomeningocele, comorbidities

## Abstract

Objectives: To investigate the relationship between the defects and symptoms caused by myelomeningocele (MMC) and quality of life. Design: A cross-sectional questionnaire survey. Methods: The subjects were 52 parents of children with MMC. Structured questionnaires were used: the Pediatric Quality of Life Inventory 4.0 Generic Core Scales (PedsQL^TM^ 4.0) and a Study-Specific Questionnaire (SSQ). Results: The overall PedsQL^TM^ 4.0 score was 56.4 (SD ± 14.7). A strong, significant negative correlation was found between the children’s age and emotional functioning. Functioning in this area deteriorated with age by a mean of 1.5 points per year of age. Children with no hydrocephalus functioned significantly better than those with this defect in the physical, social, and school areas (*p* < 0.05). Foot deformities significantly (*p* = 0.033) adversely affected the children’s physical functioning. Living in a single-parent family had no statistically significant impact on functioning in any of the areas analyzed (*p* > 0.05). Conclusion: Understanding the QoL of children with MMC and identifying its determinants may help in planning interventions to minimize the adverse impact of the illness.

## 1. Introduction

Myelomeningocele is the most common congenital neural tube defect [1,2]. Its root cause is a defect occurring around the fourth week of pregnancy in the process of neural tube formation in the embryonic ectoderm, in which the tube does not fully close on the dorsal side. In consequence, an opening remains in the spinal canal through which the meninges and cord may herniate [3,4]. The incidence of myelomeningocele is approximately 0.6 cases per 1000 live births [5], but it differs by country. It is slightly more common in girls than in boys. Neutral tube defects (NTDs) have multiple causal factors, both exogenous and genetic. Kase et al. [6] found four single nucleotide polymorphisms (SNPs) in the SOD1 gene (rs202446, rs202447, rs4816405, and rs2070424) and one SNP in SOD2 (rs5746105) associated with the risk of developing myelomeningocele in the population. Hydrocephalus and the Chiari II malformation (CM II) are the most frequently observed anomalies associated with MMC [7], resulting in delayed psychomotor development. CM II may obstruct the flow of cerebrospinal fluid, which may contribute to the development of hydrocephalus and the need for shunt diversion [8]. Only prenatal surgical treatment, known as in utero MMC repair, can protect the fetus from developing CM II. The most common myelomeningocele sites are the lumbar (60–70%) [9] or lumbosacral (46.6%) segments of the spine. This defect of the spinal cord results in neural pathway damage and lower limb and lesser pelvis dysfunctions, such as muscle paralysis or paresis, sensory loss, weak or absent reflexes, and anal sphincter and urinary bladder dysfunctions [10]. The defect has both physical and psychological consequences that last throughout the patient’s lifetime. Motor development, both in terms of fine and gross motor skills, is delayed, and impaired visual motor integration interferes with the acquisition of knowledge and experience from one’s environment and during education. Myelomeningocele treatment does not end once the patient is discharged from hospital after surgery. Surgical treatment must be followed by comprehensive specialist care, which both enables early identification of complications and associated pathologies and reduces the negative impact of the condition on the functioning of the whole family.

Quality of life (QoL) assessment has gained particular importance in the context of the World Health Organization (WHO)’s definition of health, which is not just the absence of disease or dysfunction but a state of complete physical, mental, and social wellbeing [11]. This notion plays a special role in cases of chronic illness, which places a burden on all aspects of being: physical, psychological, emotional, and social, as well as family and professional life. It is a measure of the perceived impact that one’s chronic condition has on one’s physical and psychosocial functioning [12]. Despite advances in the treatment of congenital disorders, from infancy into adolescence, long-term complications of the various therapies and psychological issues (discomfort, interference with life plans) may adversely affect the QoL of children and adolescents. 

The purpose of this paper is to investigate the relationship between the defects and symptoms caused by myelomeningocele (MMC) and the quality of life of the children affected.

## 2. Materials and Methods 

### 2.1. Study Design 

A descriptive, correlational, cross-sectional study was performed in the first quarter of 2019 and included the parents of children with myelomeningocele provided with specialist care via the Association of Patients with Myelomeningocele in Poland. The sample contained parent proxy-report data on 52 children aged 2 to 18 years (Figure 1). Surveys were completed by parents (one guardian for one child; depending on the will of the caregivers, father or mother) during the families’ stay at the hospital and the follow-up visit in the regional rehabilitation centers. All respondents received a hard copy of the questionnaire to complete with an information sheet providing instructions. In total, 58 people were invited to participate; 2 declined and 4 did not return their questionnaires. Ultimately, 52 correctly completed surveys were included in the analysis, of which 75% were filled out by female participants (*n* = 39) and 25% by male participants (*n* = 13). Before the start of the study, all parents were advised that participation would be voluntary and anonymous.

The study was conducted according to the guidelines of the Declaration of Helsinki and approved by the Institutional Review Board (or Ethics Committee) of Wroclaw Medical University (protocol code KB 539/2019 and 14 June 2019).

The inclusion criteria were defined as follows: diagnosis of MCC; a declaration that the respondent was the patient’s guardian and lived with the patient; no diagnosed mental illness in the respondent; written informed consent to participate in the study; surgically corrected defect in the patient. The exclusion criteria were as follows: lack of written consent; incomplete survey. 

The research instruments included our own survey questionnaire, the study -specific questionnaire (SSQ), and one standardized instrument, the Pediatric Quality of Life Inventory (PedsQL™) 4.0 Generic Core Scales [13,14,15,16,17].

The SSQ, featuring family, demographic, and medical questions, was completed by participants. Clinical data, such as the type of MMC, the age of gestation, treatment used, associated anomalies, and number of admissions since treatment, were obtained from medical records. The SSQ also included the sociodemographic data of participants (e.g., age, sex, residence, income) and disease-related data (e.g., psychological evaluation).

The PedsQL is a modular instrument designed to measures health in four domains: physical, emotional, social, and school functioning. It comprises 23 items to measure the QoL over the previous month in pre-school children (5–7 years old), primary school children (8–12), and adolescents (13–18). Results are recorded on a five-item Likert scale, where 0 stands for “never a problem” and 4 stands for “almost always a problem”. Responses are reverse-scored and linearly transformed to a 0–100 scale (0 = 100, 1 = 75, 2 = 50, 3 = 25, 4 = 0), where 100 indicates the best QoL. The total score is the sum of the average scores from each subscale. Lower scores indicate a poorer QoL.

### 2.2. Data Analysis

The collected data were used in statistical analyses, which included the following stages:Qualitative variables (e.g., child’s sex) were cross-tabulated as numbers (*n*) and percentages (%), and associations between pairs of variables were evaluated using Pearson’s chi-squared test;For quantitative variables (e.g., age), means (M), standard deviations (SD), and the median (Me), lower quartile (Q1), upper quartile (Q3), and extreme (Min and Max) values were calculated;For all quantitative variables, distribution normality was verified using the Shapiro–Wilk test;The significance of differences between mean values of quantitative variables with a normal distribution and homogeneous variances in two independent groups (e.g., comorbidity: present or absent) was verified using Student’s t-test. For larger numbers of groups (e.g., areas of functioning), single-factor analysis of variance (ANOVA) was used;The strength and direction of linear correlations between two continuous variables was determined by regression analysis. Pearson’s linear correlation coefficient (*r*) was calculated and, if significant (*p* < 0.05; *r* ≠ 0), i.e., if the regression coefficient significantly differed from 0, regression equation coefficients (bi) were estimated. Regression coefficients were estimated using the least square method;Calculations were performed using Statistica v. 12.5 software and Excel spreadsheets.

## 3. Results

### 3.1. Baseline Characteristics

Table 1 summarizes the characteristics of the group studied.

In total, 21.2% of the children studied were male (*n* = 11) and 78.8% were female (*n* = 41). The breakdown by age was as follows: 5–7 years old—32.7%, 2–4 years old—28.8%, 8–12 years old—25%, 13–18 years old—13.5%, mean age—7.5 ± 10.0. In most children (62%), MMC was located in the lumbar-sacral area, in 33% it was in the thoracolumbar area, and in 5% it was lumbosacral. Hydrocephalus was found in 43% of patients, and 23% of patients had Arnold-Chiari syndrome. Based on the psychological questionnaire, below-average intellectual development was found in 31% of the children, mild intellectual disability in 7%, moderate intellectual disability in 6%, and severe intellectual disability in 5%. More than half of the patients used a wheelchair, and 42.3% reported pain. Living in a single-parent family had no statistically significant impact on functioning in any of the areas analyzed, including EF (*p* > 0.05). The mean number of hospitalizations was significantly lower among children whose parents were professionally active than among those whose parents did not work professionally (*p* = 0.044). Mothers were more likely than fathers to give up working (28 vs. 4, *p* = 0.003).

### 3.2. Analysis of QoL Measured Using the PedsQL^TM^ 4.0 Generic Core Questionnaire 

The impact of the child’s illness on their physical functioning (PF), emotional functioning (EF), social functioning (SF), and school/preschool/nursery (role) functioning (RF) was evaluated, and the results are shown in Table 2. The overall PedsQL Generic Core score was 56.4 (SD ± 14.7) (Table 2).

PF was significantly poorer than EF, SF, and RF. Physical functioning, including movement and activity (walking, running, participating in sports or exercise), performing self-care, and symptoms such as fatigue and pain, was rated the poorest (total score: 41.8, SD = 20.1). The best-rated aspect was emotional functioning (total score: 63.9, SD = 14.5), including such factors as feeling afraid, sad, or angry and having trouble sleeping. Multiple comparison results are shown in Table 3. The differences between EF and SF and between EF and RF were the only ones found to be not significant (*p* > 0.05). 

### 3.3. The Impact of Demographic Characteristics on QoL 

A strong, statistically significant negative correlation was found between the children’s age and emotional functioning (Table 4). Functioning in this area deteriorated with age by a mean of 1.5 points per year of age. Residence had no statistically significant impact on the children’s functioning in any of the analyzed areas (*p* > 0.05; PF, *p* = 0.313; EF, *p* = 0.756; SF, *p* = 0.937; RF, *p* = 0.352). Interestingly, although gestational age had no statistically significant impact on the children’s functioning in any of the analyzed areas (*p* > 0.05), the poorest QoL was found in terms of PF (total score: 39.2, SD ± 17.9).

### 3.4. QoL in Children with MMC

When analyzing PedsQL scores in the context of associated comorbidities, low PF scores (total: 40.8, SD ± 20.3) were found compared to other areas such as EF (total: 64.4, SD ± 14.7) and SF (total: 59.0, SD ± 20.4). Children with no hydrocephalus functioned significantly better than those with this defect in the physical, social, and school/preschool areas (*p* < 0.05). Overall functioning was also better among children without hydrocephalus (*p* < 0.01, Table 5).

The presence of a neurogenic bladder had no statistically significant impact on functioning in any of the areas analyzed (*p* > 0.05). Despite the lack of statistical significance, the data indicate that the children’s functioning was the poorest in the physical area (total score: 31.6, SD ± 18.6). The presence of a neurogenic bowel significantly (*p* = 0.040) affected the children’s physical functioning (Table 6).

Depending on the level of the spinal cord defects, neurosegmental lesion can result in paresis of the lower limbs, as well as deformity of the hip, knee, and foot. The presence of superficial and deep sensation loss had no statistically significant impact on functioning in any of the areas analyzed (*p* > 0.05). Children with lower extremity paralysis and paresis had significantly (*p* = 0.002) worse physical functioning (Table 7). Furthermore, their overall functioning score was lower (total: 32.2, SD ± 15.8). Importantly, foot deformities significantly (*p* = 0.033) deteriorated the physical functioning of children (Table 8).

Foot deformities significantly (*p* = 0.033) adversely affected the children’s PF (Table 8).

## 4. Discussion

Congenital myelomeningocele may affect patients’ QoL in specific ways. Advances in treatment and perioperative care have significantly reduced surgical risk, improved long-term outcomes, and prolonged patients’ lives. Still, one may wonder whether surgical success and clinical outcomes correlate with patients’ self-reported wellbeing and quality of life.

The present findings indicate a considerable deficit in QoL among children with myelomeningocele. This is consistent with other studies which relied both on PedsQL scales and on other instruments. These other reports confirm that the QoL of patients with myelomeningocele is poorer than that of healthy children [18,19,20,21,22,23,24]. Furthermore, both in the present study and in those by other authors, the lowest-scored area was physical functioning and the highest-scored one was emotional functioning [25].

The proposition that many of the characteristics of a child’s physical condition are associated with QoL should be investigated further to determine whether these effects simply reflect the general severity of illness of a child or the specific adverse effect of a particular function.

Many factors have been investigated for possible effects on the QoL of patients with MMC. Some researchers have noted that poor QoL is clinically related to age, ambulatory ability, or family functioning [26,27]. Meanwhile, other studies [28,29] have noted that QoL in MMC is unrelated to age, functional ability, sex, or lesion level. Our investigation found a strong, statistically significant correlation between the children’s age and a loss of emotional functioning (1.5 points per year of age). Further analysis of the effects of aging on MMC patients may be warranted to examine this finding. 

Studies have confirmed the association between the severity of the condition and QoL. In our analysis, the percentage of patients with hydrocephalus was 43%, which is well below the usual range of 70–86% [30]. The explanation for this is that other studies included all spinal dysraphisms, while our sample only focused on myelomeningocele. The presence of hydrocephalus was associated with poorer QoL in children with MMC. These data are consistent with other studies showing an association between factors other than disability and children’s QoL [31,32]. The effects of hydrocephalus on children’s functioning vary considerably across children and depend on the areas of the brain most affected.

What should be noted and applied in subsequent studies is that Kulkarni et al. [33] developed a measurement of QoL, the Hydrocephalus Outcome Questionnaire (HOQ). In a study using the HOQ, Kulkarni et al. demonstrated that shunt infection and other shunt-related complications were predictors of poor quality of life.

Renal deterioration is a major cause of mortality for MMC patients of all ages. It results in chronic elevated bladder pressure with filling and incomplete emptying [34]. Clayton et al. [35] found that urinary incontinence was a significant cause of quality of life deterioration and limited social independence. However, in our sample of patients with neurogenic bladders there was no statistically significant impact on functioning in any of the QoL areas analyzed, while a neurogenic bowel was found to significantly affect children’s physical functioning. 

Most children with MMC present with foot and ankle deformities. The cosmetic appearance caused by these deformities is not the only problem, as children may develop skin irritation and experience difficulties in ambulation. This is consistent with our research, where foot deformities significantly deteriorated the physical functioning of children.

An illness of one family member affects the entire family, disrupting the balance of the family system. Families are inherently resilient and, in the face of adversity, work together to regain stability [36,37]. However, as stated by Solomon et al. [38], having a child with a chronic disease leads to family difficulties and generates a sense of isolation, loneliness, powerlessness, and social marginalization in parents. Living with a child with a disability requires making many sacrifices every day. The stress caused by the presence of a child with MMC in the family increases the need to learn to deal with the difficult situation, something that is not easy to accomplish in addition to the normal challenges faced by parents. It has not been shown so far why some parents are better at this, while others are worse. Okurowska-Zawada et al. [39] indicated that mothers of children with MMC have a reduced perception of QoL in all domains (social, physical, emotional) compared with mothers of healthy children. Additionally, living in a single-parent family brings with it numerous difficulties, if only due to the greater psychological and physical burden and lower level of support from relatives. This can result in a lower QoL for both parents and their children. Our research did not confirm the relationship, but nevertheless it was mothers who resigned from work more often than fathers to care for their children. 

Studies on quality of life in relation to health deliver not only holistic knowledge and understanding of the consequences of the disease but also a thorough assessment of the needs of patients and their caregivers. The acceptance of the results of HRQoL studies is a breakthrough in medicine. The patient ceases to be subjected to treatment and their opinion and the assessment of the wellbeing associated with the therapy are taken into account. Therefore, researchers have started following principles of medical practice and nursing practice based on facts—evidence-based medicine (EBM) and evidence-based nursing practice (EBNP), respectively—taking into account the feelings of the patient and their preferences [40,41]. By implementing research findings in clinical practice, one can plan further interventions more effectively, e.g., by providing the patient and their family with psychological support, which in the long term may not only improve their QoL or treatment outcomes but also help physicians have a better relationship with patients and their families. The findings of this study have to be seen in light of their limitations. The first limitation is that QoL was assessed only from the parents’ perspective. Parental reports may not fully reflect the subjective experience of their children. The second limitation concerns the fact that quality of life was assessed on the basis of generic scales; in the future studies, a child-specific questionnaire should be used. 

## 5. Conclusions

Children with myelomeningocele require comprehensive long-term care, and their families must contribute to the treatment and education process. Understanding the QoL of children and adolescents born with MMC and identifying its determinants is important, as it may help in planning interventions to minimize the adverse impacts of the illness and its treatment on patients’ physical and emotional conditions and ensure their optimal functioning in the family and the society.

## Figures and Tables

**Figure 1 ijerph-18-10756-f001:**
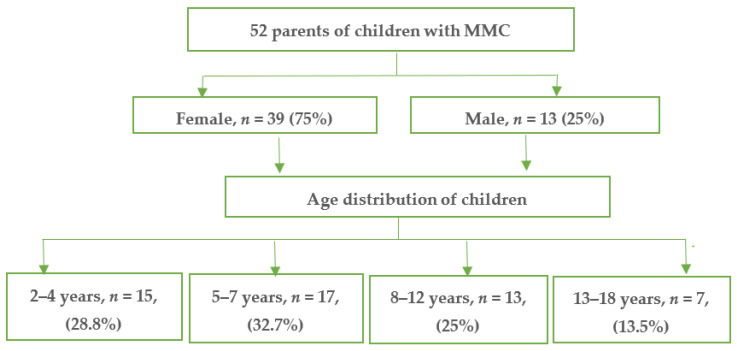
Flowchart of study population.

**Table 1 ijerph-18-10756-t001:** Characteristics of the parents and their children.

Characteristic (Variable)	Statistics	
Parent’s age (years)		
M ± SD	35.5 ± 7.2	
Me [Q1; Q3]	35 [30; 40]	
Min–Max	22–58	
Child’s age (years)		
M ± SD	7.5 ± 10.0	
Me [Q1; Q3]	7 [4; 10]	
Min–Max	1–17	
Child’s sex (*n* and %)		
Girl	41	78.8
Boy	11	21.2
Residence (*n* and %)		
Rural	25	48.1
Urban	27	51.9
Gestational age at birth (*n* and %)		
≤37 weeks	30	57.7
>37 weeks	22	42.3
Children diagnosed with associated birth defects (*n* and %)	49	94.2
Associated comorbidities (*n* and %)		
Hydrocephalus	22	42.3
Neurogenic bladder	42	80.7
Neurogenic bowel	38	73.1
Superficial and deep sensation loss	13	25.0
Lower extremity paralysis or paresis	23	44.2
Spinal deformity	21	40.4
Foot deformities (*n* and %)		
Talipes equinovarus	24	46.2
Talipes calcaneus	2	3.8
Hip dysplasia	9	17.3
Other	4	7.7
Has the child’s illness forced the parent to quit their job (*n* and %)	32	61.5
Which parent has quit their job? (*n* and %)		
Mother	28	53.8
Father	4	7.7

M, mean; Me, median; Min, minimum; Max, maximum; SD, standard deviation.

**Table 2 ijerph-18-10756-t002:** Survey results for the parents of 52 children with myelomeningocele.

Questionnaire	Descriptive Statistics		
	M ± SD	Me [Q1; Q3]	Min–Max
Physical functioning (PF)	41.8 ± 20.1	42 [25; 53]	9–100
Emotional functioning (EF)	63.9 ± 14.5	65 [50; 73]	35–100
Social functioning (SF)	58.6 ± 19.9	63 [48; 70]	10–100
School/preschool/nursery (role) functioning (RF)	72.4 ± 22.9	75 [58; 92]	8–100
Overall functioning score	56.4 ± 14.7	57 [49; 64]	27–92

M, mean; Me, median; Min, minimum; Max, maximum; SD, standard deviation.

**Table 3 ijerph-18-10756-t003:** Results of multiple comparisons between scores for the investigated areas of the children’s functioning.

Area of Functioning	PF	EF	SF	RF
	M = 41.6	M = 63.9	M = 58.6	M = 72.4
PF	×	*p* < 0.001	*p* < 0.001	*p* < 0.001
EF	*p* < 0.001	×	*p* = 0.489	*p* = 0.120
SF	*p* < 0.001	*p* = 0.489	×	*p* = 0.002
RF	*p* < 0.001	*p* = 0.120	*p* = 0.002	×

M, arithmetic mean; PF, physical functioning; EF, emotional functioning; SF, social functioning; RF, school/preschool/nursery (role) functioning.

**Table 4 ijerph-18-10756-t004:** Pearson’s correlation coefficients for children’s age and functioning.

Child’s Area of Functioning				Total Score
PF	EF	SF	RF	
*r* = −0.141	*r* = −0.445	*r* = −0.019	*r* = −0.220	*r* = −0.211
*p* = 0.320	*p* = 0.001	*p* = 0.896	*p* = 0.117	*p* = 0.133

PF, physical functioning; EF, emotional functioning; SF, social functioning; RF, role functioning.

**Table 5 ijerph-18-10756-t005:** Basic statistics (M ± SD) for children’s functioning scores, broken down by the presence or absence of hydrocephalus, with significance test results.

Child’s Area of Functioning	Hydrocephalus		*p*-Value
	Present *n* = 49	None *n* = 3	
Physical functioning (PF)	32.2 ± 17.1	48.9 ± 19.4	0.002
Emotional functioning (EF)	60.7 ± 15.2	66.3 ± 13.7	0.167
Social functioning (SF)	50.0 ± 22.0	64.8 ± 15.8	0.007
School/preschool/nursery (role) functioning (RF)	64.8 ± 27.2	78.1 ± 17.6	0.037
Total score	49.2 ± 13.8	61.6 ± 13.2	0.002

**Table 6 ijerph-18-10756-t006:** Basic statistics (*M* ± *SD*) for children’s functioning scores, broken down by the presence or absence of a neurogenic bowel, with the significance of the test results.

Child’s Area of Functioning	Neurogenic Bowel			*p*-Value
	None *n* = 10	Constipation *n* = 10	Constant Passing of Stool *n* = 32	
Physical functioning (PF)	53.1 ± 17.8	38.1 ± 20.0	32.3 ± 14.4	0.040
Emotional functioning (EF)	65.7 ± 19.6	62.6 ± 12.6	71.7 ± 5.8	0.512
Social functioning (SF)	59.3 ± 19.9	58.7 ± 18.7	53.3 ± 88.8	0.896
School/preschool (role) functioning (RF)	71.4 ± 23.5	71.2 ± 23.0	91.7 ± 14.4	0.332
Total score	60.3 ± 17.5	54.9 ± 13.6	55.5 ± 15.1	0.508

**Table 7 ijerph-18-10756-t007:** Basic statistics (M ± SD) for children’s functioning scores, broken down by the presence or absence of lower extremity paralysis or paresis, with the significance of the test results.

Child’s Area of Functioning	Lower Extremity Paralysis or Paresis		*p*-Value
	Present *N* = 23	None *N* = 29	
Physical functioning (PF)	32.3 ± 15.8	49.4 ± 20.2	0.002
Emotional functioning (EF)	60.7 ± 13.8	66.6 ± 14.7	0.147
Social functioning (SF)	52.8 ± 22.4	63.1 ± 16.7	0.064
School/preschool/nursery (role) functioning (RF)	65.9 ± 25.6	77.6 ± 19.4	0.068
Total score	50.1 ± 12.7	61.3 ± 14.4	0.005

**Table 8 ijerph-18-10756-t008:** Basic statistics (M ± SD) for children’s functioning scores, broken down by the presence or absence of foot deformities, with the significance of the test results.

Child’s Area of Functioning	Foot Deformities		*p*-Value
	Present *N* = 30	None *N* = 22	
Physical functioning (PF)	36.8 ± 14.4	48.7 ± 24.7	0.033
Emotional functioning (EF)	62.2 ± 12.4	66.4 ± 16.9	0.307
Social functioning (SF)	58.8 ± 16.2	58.2 ± 24.4	0.908
School/preschool/nursery (role) functioning (RF)	70.3 ± 24.0	75.4 ± 21.4	0.433
Total score	54.0 ± 10.7	59.6 ± 18.6	0.178

## Data Availability

The data that support the findings of this study are available from the corresponding author, upon reasonable request.

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
