# Peer review of "The Quality of Life of Children with Myelomeningocele: A Cross-Sectional Preliminary Study"

_ijerph, 2021, doi:10.3390/ijerph182010756_

Round 1
Reviewer 1 Report
This study is to explore the quality of life (QOL) of children with myelomeningocele (MMC) and the assumption predictors. Understanding the QoL of children MMC and identifying its determinants, may help in planning interventions to minimize the adverse impact of the illness.
This is an interesting study, but I have a few questions for acceptance.
・I had the impression that the introduction was long. I think it would be easier to read if it were more compact.
・Why did you choose myelomeningocele as the disease subject? There are many neurological diseases that require long term rehabilitation with medical and surgical treatment options.
・The report would be more global if it introduced the environment for child care and medical assistance in Poland and, if possible, compared the environment with other countries.
Author Response
Dear Reviewer,
Thank you very much for sending us the consensus opinion about requested revision of our manuscript entitled: The Quality of Life Of Children with Myelomeningocele—a Cross-Sectional Preliminary Study. The manuscript had been revised according to the reviewers comments and criticism. Most suggestions were accepted and incorporated into the text.

Reviewer 2 Report
Dear authours,
Some aspects must be corrected:
Abstract
Introduction should be in relation to the topic, not to the study description.
Introduction: To start defining Myelomeningocele, the alterations and associated problems and after defining the QoL...an the objetive of the study.
Methods: To include a flow diagram with the sample.
Discussion
What are the limitations? This part (6 in your manuscript, after conclusion) must be at the end of the discussion.
What therapies could be used in this population?
Re-write the conclusion in a concise way
Author Response

(The authors gave the same response as above.)

Reviewer 3 Report
The authors present a cross-sectional study of quality of life of children with myelomeningocele using the PedsQL instrument. The show a declining QOL with increasing age and an association of lower QOL with presence of shunted hydrocephalus. These observations have been made in previous studies. Nevertheless, this is a well done study.
Can the authors provide clarification in the methods section about the participants. Is it only parents who complete the instrument? Some variables appear to be reported for the parents, but others for the children. This should be made more clear.
There are far more girls than boys in the sample. This is not a typical observation in children with myelomeningocele. Do the authors have any thoughts on why this might be?
Hydrocephalus prevalence is quite low at 42%. It is more typical to see rate of hydrocephalus closer to 80% in children who have undergone post-natal closure of myelomeningocele. Do the authors have any thoughts on why this rate of hydrocephalus is so low.
Did any of these children undergo fetal myelomeningocele closure?
Author Response

(The authors gave the same response as above.)

Round 2
Reviewer 1 Report
Thank you for your revision.
The manuscript will be revised appropriately according to the reviewers' comments.
I believe that this study is worthy of acceptance.